# Evaluation and comparison of nine growth and development-based measures of pubertal timing
Ahmed Elhakeem [1,2] ✉, Monika Frysz [1,3], Ana Goncalves Soares [1,2], Joshua A. Bell[1,2], Tim J. Cole [4], Jon Heron[1,2], Laura D. Howe[1,2], Sylvain Sebert [5], Kate Tilling [1,2], Nicholas J. Timpson [1,2] & Deborah A. Lawlor [1,2,6]

## Abstract

**Background** Pubertal timing is heritable, varies between individuals, and has implications for life-course health. There are many different indicators of pubertal timing, and how they relate to each other is unclear. Our aim was to quantitatively compare nine indicators of pubertal timing.

**Methods** We used data from questionnaires and height, weight, and bone measurements from ages 7–17 y in a population-based cohort of 4267 females and 4251 males to compare nine growth and development-based indicators of pubertal timing. We summarise age of each indicator, their phenotypic and genetic correlations, and how they relate to established genetic risk score (GRS) for puberty timing, and phenotypic childhood body composition measures.

**Results** We show that pubic hair in males (mean: 12.6 y) and breasts in females (11.5 y) are early indicators of puberty, and voice breaking (14.2 y) and menarche (12.7 y) are late indicators however, there is substantial variation between individuals in pubertal age. All indicators show evidence of positive phenotypic intercorrelations (e.g., r = 0.49: male genitalia and pubic hair ages), and positive genetic intercorrelations. An age at menarche GRS positively associates with all other pubertal age indicators (e.g., difference in female age at peak height velocity per SD higher GRS: 0.24 y, 95%CI: 0.21 to 0.26), as does an age at voice breaking GRS (e.g., difference in age at male axillary hair: 0.11 y, 0.07 to 0.15). Higher childhood fat mass and lean mass associated with earlier puberty timing.

**Conclusions** Our findings provide insights into the measurements of the timing of pubertal growth and development and illustrate value of various pubertal timing indicators in life-course research.

## Plain language summary

Age of puberty varies between individuals and can affect a person's future health. We obtained information from 8500 British children as they progressed through puberty. We compared nine measures of pubertal timing. We found that the appearance of pubic hair in boys and breasts in girls are early indicators of puberty, and that voice change and onset of menstruation are late indicators. However, there was also substantial variability between individuals in age of puberty. All puberty measures were correlated with each other and related to an individual's adult body mass index, as well as to their childhood muscle and fat mass. Our findings are useful information for health care workers and researchers who are interested in assessing and studying puberty.

Puberty is a milestone in human development that involves rapid transformations in anatomy, physiology, and behaviour. Its central feature is neuroendocrine transformation of processes regulating reproductive physiology via a reactivation of the hypothalamic-pituitary-gonadal (HPG) axis, leading to the onset of adult reproductive capacity[1,2]. Reactivation of the HPG axis produces numerous observable downstream consequences including production of gonadal steroids, a pubertal growth spurt,

development of secondary sexual characteristics, onset of menstruation in females, and appearance of facial hair and voice change in males[2,3]. The sequence in which the observable changes appear is thought to mirror elevation of steroid levels, with all changes occurring earlier in females than males[2].

There is substantial variation in the age of puberty between children[4,5], which is attributable to genetic as well as non-genetic factors, such as

[1]MRC Integrative Epidemiology Unit at the University of Bristol, Bristol, UK. [2]Population Health Science, Bristol Medical School, University of Bristol, Bristol, UK. [3]Musculoskeletal Research Unit, Translational Health Sciences, Bristol Medical School, University of Bristol, Bristol, UK. [4]UCL Great Ormond Street Institute of Child Health, London, UK. [5]Research Unit of Population Health, University of Oulu, Oulu, Finland. [6]NIHR Bristol Biomedical Research Centre, Bristol, UK. ✉e-mail: a.elhakeem@bristol.ac.uk

nutrition[6–9]. Understanding the determinants of variation in pubertal timing between individuals is important given its relation to reproductive capability and social and health implications, including the risk of some cancers[8–16]. Within an individual, there is variation in the timing of different maturation processes (e.g., skeletal, and sexual maturation), and in the timing of related structures within a maturational process (e.g., within the sexual maturation process, pubic hair and genitalia can have different levels of maturity)[17]. Various approaches and indicators have been used by studies to measure puberty timing[17–21], which have included self/parent-reported age at menarche and voice change in females and males, respectively, and longitudinally modelled age at peak height velocity in both. As no single measure can capture all maturational processes during puberty, detailed, systematic analysis of anthropometric and developmental measures of puberty timing, including in both sexes, can help reveal their value for life course research, and identify which measure is best for exploring the causes and consequences of pubertal timing, and what might be done to mitigate those effects.

The aim of this study was to evaluate and compare multiple measures of pubertal timing. We used a UK birth cohort—the Avon Longitudinal Study of Parents and Children (ALSPAC)[22–24]—where offspring have been prospectively assessed since birth with extensive biomedical data collections that included repeated assessments of height, weight, and bone in research clinics, and repeated assessments of pubertal development. Importantly, assessments began at age 7 years, i.e., before onset of puberty in most children. We derive nine anthropometric and development-based measures of pubertal age in >8500 females and males and describe the timing and chronological sequence of pubertal growth and development, the phenotypic and genetic correlations between measures of pubertal age, and how each pubertal age measure relates to genetic risk scores (GRSs) for pubertal timing and adiposity, and phenotypic measurements of childhood body composition. We identify early and late indicators of puberty and find that all pubertal age measures are interrelated. We show that pubertal age measures are related to GRSs for pubertal timing and adiposity and to phenotypic measurements of childhood fat mass and lean mass.

## Methods

This study was conducted using data from the ALSPAC cohort. A pre-specified analysis plan for this study is available at https://osf.io/3qndg/[25].

### Cohort description

ALSPAC is a multigenerational prospective birth cohort study that recruited pregnant women residing within the catchment area of three National Health Service authorities in southwest England with an expected date of delivery between April 1991 and December 1992[22–24]. The initial number of pregnancies enrolled was 14,541. Of these initial pregnancies, there was a total of 14,676 fetuses, resulting in 14,062 live births and 13,988 children who were alive at 1 year of age. When children were ~7 years old, an attempt was made to bolster the initial sample with eligible new cases. Total sample size for analyses using data collected after age 7 years was 15,447 pregnancies, and 15,658 offspring. Of these 14,901 were alive at 1 year of age. Detailed data have been collected from offspring and parents by questionnaires, data extraction from medical records, data linkage to health records, and dedicated clinic assessments.

ALSPAC participants provided written informed consent for all measurements. Parents gave informed consent for children aged under 18 years and the children were also invited to give assent, with no measurements were taken from the children if they refused. Ethical approval for the ALSPAC study was obtained from the ALSPAC Law and Ethics Committee and the Local Research Ethics Committees (Bristol and Weston Health Authority, Southmead Health Authority, Frenchay Health Authority, United Bristol Healthcare Trust, North Bristol Trust, Weston Area Health Trust, Central & South Bristol Research Ethics Committee, North Somerset Research Ethics Committee, National Research Ethics Service Committee South West). Consent for biological samples has been collected in accordance with the Human Tissue Act (2004). Details of all available data can be found in the ALSPAC study website which includes a fully searchable data dictionary and variable search tool (http://www.bristol.ac.uk/alspac/researchers/our-data/).

### Puberty data collection from research clinics and questionnaires

Data used to derive indicator-based pubertal ages were collected prospectively using nine repeated research clinic assessments and nine puberty-specific questionnaires. Figure 1 summarises the observed data from these clinic assessments and questionnaires, and Supplementary Table 1 and Supplementary Table 2 provide more information.

All participants were invited to attend nine repeated research clinic examinations from ages 7–17 years where their height (in cm) and weight (in kg) were measured. In five of the clinics (ages 9–17 years), all participants underwent whole-body Dual-energy X-ray Absorptiometry (DXA) scans from which total-body (less head) bone mineral content (BMC; in grams) was extracted. Exact age in months at attending each research clinic assessment was recorded.

Questionnaires on pubertal development (the 'Growing and Changing Questionnaire') were mailed to all participants on nine occasions from ages 8 to 17 years. Questionnaires could be answered by either the parent or guardian, child, or a combination; over 70% of the first five questionnaires were completed with help from a parent or guardian whereas the last four were mostly completed by the child alone (Supplementary Table 3). Each questionnaire collected data on the five Tanner stages of pubic hair, breasts (girls), and genitalia (boys) development using line drawings representing each stage with accompanying description (Supplementary Note). Each questionnaire collected data on onset of menstruation in girls, and all except the first questionnaire collected data on change in voice (boys). The last seven questionnaires (ages 10–17 years) gathered data on the development of axillary hair. Exact age in months at completing each puberty questionnaire was recorded.

### Genotyping and imputation

Children were genotyped using the Illumina HumanHap550 quad chip genotyping platform (Illumina) by 23andMe subcontracting the Wellcome Trust Sanger Institute (Cambridge, UK) and the Laboratory Corporation of America (Burlington, NC, USA). Raw genome-wide data were subjected to standard quality control methods. Individuals were excluded based on sex mismatches, minimal or excessive heterozygosity, disproportionate missingness (>3%), and insufficient sample replication (identity by descent (IBD) < 0.8). Individuals of non-European ancestry were removed because source GWAS (described below) for puberty measures were conducted primarily in European populations. Single nucleotide polymorphisms (SNPs) with minor allele frequency <1%, call rate <95%, or evidence for violations of Hardy-Weinberg equilibrium ($P < 5 \times 10^{-7}$) were removed. Cryptic relatedness was measured as proportion of IBD > 0.1. Related individuals that passed quality control thresholds were retained in subsequent phasing and imputation.

In total, 9115 children and 500,527 SNPs passed quality control filters. Of these, 477,482 SNP genotypes in common between the sample of ALSPAC children and mothers were combined for imputation to the Haplotype Reference Consortium (HRCr1.1, 2016) panel. SNPs with genotype missingness >1% (11,396 SNPs) were removed prior to imputation. A further 321 subjects were removed due to ID mismatches. HRC panel was phased using ShapeIt (v2.r644) which utilizes relatedness during phasing, and imputation was performed using the Michigan imputation server. This resulted in 8237 children with genotype data after exclusion of related subjects using cryptic relatedness measures described previously.

### GRSs for female and male pubertal timing, and adulthood and childhood BMI

Four separate GRSs were created using genome-wide significant SNPs from four European ancestry GWAS meta-analyses on reported age at menarche[8] and age at voice breaking[9], and measured BMI in adulthood (mostly middle-aged adults)[26] and childhood (age range from 3 to 10 years)[27]. Scores were calculated using 351 SNPs associated with age at menarche[8], 73 SNPs

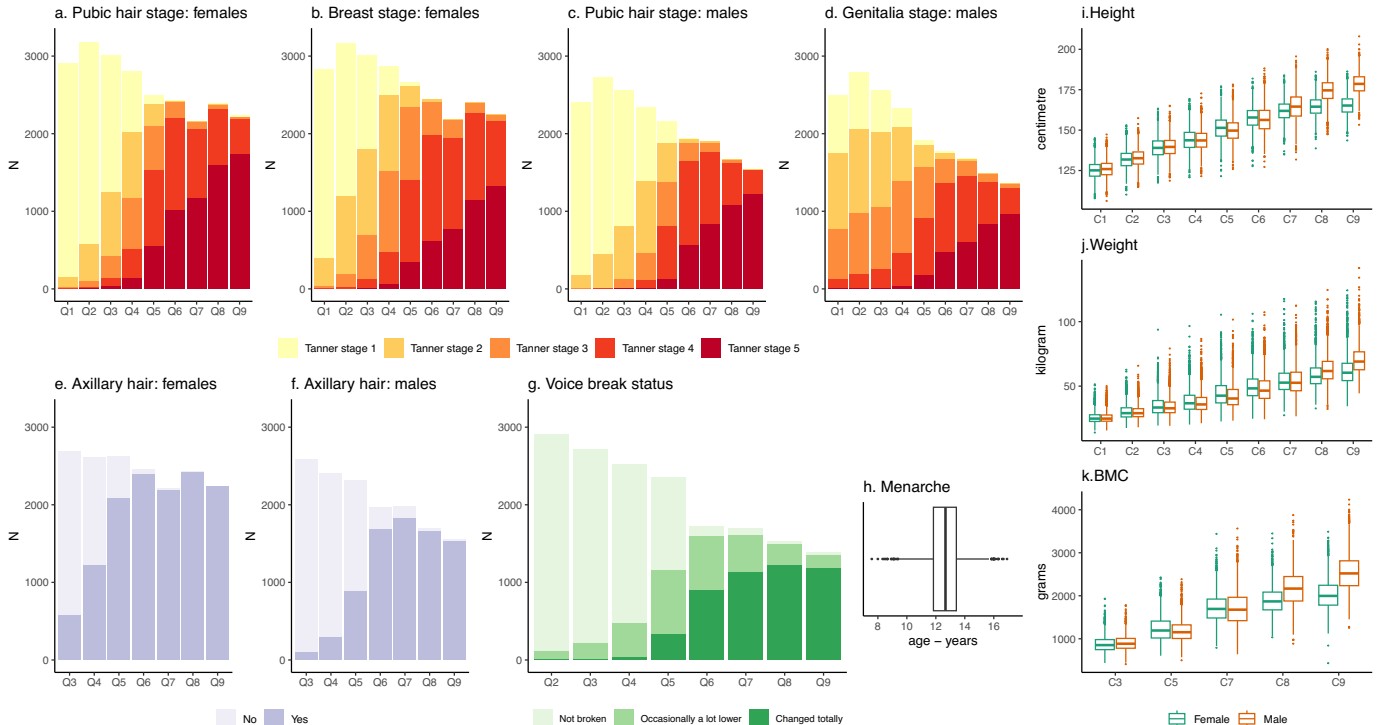

**Fig. 1 | Longitudinal pubertal growth and development data that were used to derive nine indicator-based measures of pubertal age.** Figure shows number of study participants in each pubic hair (**a, c**), breast (**b**), and genitalia Tanner stage (**d**), axillary hair groups (**e, f**), and voice breaking group (**g**) at each puberty questionnaire (Q), the distribution of age at menarche (**h**), and the distribution of height (**i**), weight (**j**), and bone mineral content (BMC) (**k**) at each research clinic (C). The total numbers of study participants from which data is plotted are 4276 females and 4251 males. The number of study participants and the age at completing each puberty questionnaire and clinic assessment is shown in Supplementary Table 2.

associated with age at voice breaking[9], 95 SNPs associated with adulthood BMI[26] (2/97 SNPs were not available in ALSPAC), and 15 SNPs associated with childhood BMI[27]. The scores were constructed by multiplying the number of effect alleles (or probability of effect alleles if imputed) at each SNP (0, 1, or 2) by its weighting, summing them, and dividing by the total number of SNPs used, and reflect the average per-SNP effect on their respective trait (age at menarche, age at voice breaking, adulthood BMI, or childhood BMI). All scores were standardised (to mean=0 and SD = 1) prior to analysis (Supplemental Fig. 1).

**Childhood body composition measurements and confounders**
Pre-pubertal fat mass index (total body fat mass divided by height squared) and lean mass index (total body lean mass divided by height squared), both in units of kg/m², were derived from DXA scans performed at mean age 9.9 years and were used to examine associations of childhood body composition with pubertal timing. DXA scans were performed using a Lunar Prodigy scanner (Lunar Radiation Corp) and were analysed according to the manufacturer's standard scanning software and positioning protocols. Scans were reanalysed as necessary to ensure optimal placement of borders between adjacent subregions, and scans with anomalies were excluded. Exact age in months when scan was performed was recorded. Fat mass and lean mass indices were standardised (to mean = 0 and SD = 1) prior to examining association with the derived indicator-based pubertal ages.

Maternal education, maternal early pregnancy BMI and early pregnancy smoking, maternal age at birth, parity, and child's diet were identified as factors that could plausibly influence both child body composition and pubertal timing and were selected to be included as confounder adjustment when examining associations of childhood fat mass and lean mass indices with pubertal timing. Maternal confounders were reported using questionnaires during pregnancy (maternal BMI was calculated from reported height and weight). Child's diet was based on daily energy

intake (in kilojoules per day) and derived from food frequency questionnaires completed by the parent when the child was aged 7 years. Confounders were reported in questionnaires during pregnancy for maternal factors.

**Statistics and reproducibility**
Nine indicator-based pubertal timing (i.e., age) measures were derived: two in females only (age at menarche and age in Tanner breast stage 3), two in males only (age at voice breaking and age in Tanner genitalia stage 3), and five in both females and males (age at peak BMC, height, and weight velocity, age in Tanner pubic hair stage 3, and age at axillary hair).

Estimated pubertal ages were analysed in months for all measures and presented in years to aid interpretation. All analyses were restricted to White ethnicity individuals (>95% of all participants) to enable consistency across phenotypic and genetic analyses. Analyses were performed in R version 4.02 (R Project for Statistical Computing).

Age at menarche was calculated as the first reported age at onset of menstruation. Pubertal age for all other measures was derived using the SITAR (Super Imposition by Translation And Rotation) method of growth curve analysis[28,29]. SITAR is a shape invariant nonlinear mixed effects model that fits a single (mean) natural spline growth curve in the study sample and tailors it (using random effects) to define how individual growth curves differ from the mean curve. SITAR usually has up to three random effects that describe the *size*, *timing*, and *intensity* of individual growth relative to the mean growth curve. *Size* adjusts for differences in growth and geometrically reflects up or down shifts in the mean curve, *timing* adjusts for differences in the timing of peak growth and geometrically reflects left to right shifts in the mean curve, and *intensity* adjusts for the duration of the growth spurt and geometrically corresponds to shrinking or stretching of the age scale (which rotates the mean curve)[28]. A recent addition to the SITAR software allows a

fourth 'post-growth' random effect to be fitted which extends SITAR to model variability in the adult slope of the growth curve to allow post-pubertal growth rate to vary between individuals[30].

Height was modelled using the standard SITAR approach with three random effects. Weight and BMC were modelled using SITAR with all four random effects to allow for variation in growth post-puberty. Tanner stages for pubic hair, breast, and genitalia development, and voice breaking, and axillary hair were modelled using SITAR with up to two random effects for *timing* and *intensity*[31]. This reduced SITAR model (i.e., without the *size* random effect) was used as all individuals are measured on the same 5-point scale (or 3 for voice breaking, and 2 for axillary hair), and so their position on the scale at any particular time depends purely on their developmental age at that time, taking into account their *timing* and *intensity* effects.

SITAR models were fitted separately in males and females with at least one outcome measurement. The best fitting models were identified by comparing models with 2 to 5 knots (placed at quantiles of the age distribution) in the mean spline curve and inspecting the fitted mean curves and the Bayesian information criterion (BIC) values for each model (Supplementary Table 4, Supplementary Figs. 2 and 3). Covariances for the random effects were modelled (Supplementary Table 5). Indicator-based pubertal age was estimated using the *timing* random effect from each SITAR model and represent age at peak growth velocity for height, weight and BMC, age in Tanner stage 3 of pubic hair, breast (females only) and genitalia (males only) development, and age at voice breaking (males only) and axillary hair appearance. Because regression modelling can allow for measurement error, inconsistent responses (i.e., reporting a developmental stage that was lower than that reported in a previous questionnaire) were included in the analysis, except for inconsistent responses in voice breaking which were removing prior to modelling due to convergence issues. Lastly, we did a sensitivity analysis to examine the effect of outliers on anthropometric pubertal age estimates by refitting SITAR models for height, weight, and BMC and re-estimating ages at peak velocity after removing conventional putative outliers (+/−5 SD).

The timing of pubertal indicators was summarised by calculating the mean age, and variation between individuals around the average age was summarised by calculating SD. Bivariate scatterplots and pairwise phenotypic Pearson correlations were used to examine interrelationships between pubertal age measures.

Linkage disequilibrium score regression (LDSR) was used to estimate genetic correlations between pubertal age measures, both within and between sex, using full GWAS summary statistics[32]. Summary data were obtained from a published GWAS for age at menarche[8] ($n = 252,000$) and were generated in ALSPAC (coded in years) for all other measures (including voice breaking because full summary data were not available from the GWAS on age at voice breaking[9]). In ALSPAC, linear regression was used to run GWAS in BOLT-LMM[33] (without adjustment for principal components as all participants were from a small geographically defined region, with 96% of parents reporting they were White British). A reference map from BOLT-LMM was used to interpolate genetic map coordinates from each SNP physical (base pair) position. Reference LD scores from BOLT-LMM appropriate for the analysis of European-ancestry samples were used to calibrate BOLT-LMM. LD scores were matched to SNPs by base pair coordinate. GWAS was performed separately for male and female pubertal age measures, and results for shared measures (i.e., height, weight, BMC, pubic hair, and axillary hair) were meta-analysed using GWAMA[34]. ALSPAC GWAS sample sizes ranged from 3109 (age at peak BMC velocity in males) to 6782 (age at peak height velocity in females and males combined).

To evaluate the usefulness of our nine derived pubertal age measures in respect to the strength of their associations with genetic predisposition to pubertal timing and BMI, we used separate univariable linear regression models to examine associations of four standardised GRSs that were constructed from published genome-wide significant SNPs for female and male pubertal timing[8,9] and adulthood and childhood BMI[26,27] with each pubertal age measure.

Effect of pre-pubertal body composition in terms of DXA-derived fat mass and lean mass indices (at age 10 years) on pubertal age measures was examined in separate multivariable linear regression models adjusted for exact age at measurement of fat mass and lean mass, and confounders (maternal age at birth, maternal education, parity, maternal early pregnancy BMI, maternal pregnancy smoking, and childhood dietary intake). DXA measures recorded after the age of puberty were removed. Fat mass and lean mass indices were coded in age- and sex-specific SD units (mean = 0 and SD = 1).

### Reporting summary
Further information on research design is available in the Nature Portfolio Reporting Summary linked to this article.

## Results
Indicator-based pubertal age was estimated for up to 4267 females and 4251 males who had completed at least one of up to nine repeated research clinic assessments where height, weight and BMC were recorded; or at least one of up to nine repeated puberty questionnaires where menarche, Tanner stages, and axillary hair and voice breaking status were reported. When compared with those included in estimation of pubertal age, those excluded due to missing data on all clinic and questionnaires assessments had younger maternal age at birth, lower maternal education, higher prevalence of maternal pregnancy smoking, mothers who were likely to have had previous pregnancies resulting in live birth, similar maternal pre-pregnancy BMI, and somewhat higher childhood energy intake (Supplementary Table 6).

### Timing of pubertal growth and development
Mean age of pubertal indicators in females varied across measures from 11.5 years for age in Tanner breast stage 3 to 12.7 years for age at menarche (average of 1.2 years from mean age of earliest to latest measure), and in males from 12.6 years for age in Tanner pubic hair stage 3 to 14.2 years for age at voice break (average of 1.6 years from mean age of earliest to latest measure) (Fig. 2). The largest gap between the mean ages of consecutive measures was 0.3 years for females (from Tanner pubic hair stage 3 to axillary hair, and from peak BMC velocity to menarche) and 0.7 years for males (from Tanner genitalia stage 3 to axillary hair). Mean age of pubertal indicators was younger in females than males for all five measures common to both sexes, e.g., 11.8 years versus 13.5 years for age at peak height velocity (Fig. 2).

There was considerable variability between individuals in the timing of pubertal indicators, e.g., in females, the SD around mean ages ranged from 0.8 (peak height velocity) to 1.2 years (menarche), and in males, from 0.7 years (peak BMC velocity) to 1.2 years (Tanner genitalia stage 3 and peak weight velocity) (Fig. 2). Moreover, age in Tanner breast stage 3 occurred first in 38.5% of females and age at menarche was last in 41.8% and likewise, age in Tanner pubic hair stage 3 occurred first in 36.4% of males and age at voice break was last in 45.8% of males. Removing outliers had minimal impact on the estimated age at peak height, weight, and BMC velocity (the number (%) of observations removed in females and males were 99 (0.4%) and 633 (2.4%) for height, 119 (0.4%) and 238 (0.9%) for weight, and 6 (0.0004%) and 42 (0.4%) for BMC. Following removal of outliers, mean (and SD) ages at peak height, weight, and BMC velocity, respectively, were 11.5 (1.2), 11.8 (1.2), and 12.5 (1.0) years in females and 13.4 (1.1), 13.8 (1.3), and 13.2 (1.3) years in males.

### Phenotypic correlations between indicator-based pubertal age measures
Pair-wise phenotypic (Pearson) correlation analyses identified positive, generally moderate strength, correlations between all pubertal age measures, with mainly stronger correlations in females (Fig. 3) than males (Fig. 4). In females, correlations ranged from 0.28 (between age at axillary hair and age at peak weight velocity) to 0.76 (age at menarche and age at peak height velocity). In males, correlations were from 0.19 (between age in Tanner genitalia stage 3 and age at peak weight velocity) to 0.77 (age at peak height velocity and peak BMC velocity).

**Fig. 2 | Timing of pubertal growth and development.** Figure summarises the timing (age in years) of pubertal age measures in females (**a**) and males (**b**), estimated using mixed effects models for each indicator except for menarche (as it was directly calculated). Black points are mean age and horizontal black bars represent the mean $+/- 1$ SD. Grey points are the ages for each study participant. Pubertal age measures are arranged by chronological sequence from youngest to oldest (mean) age.

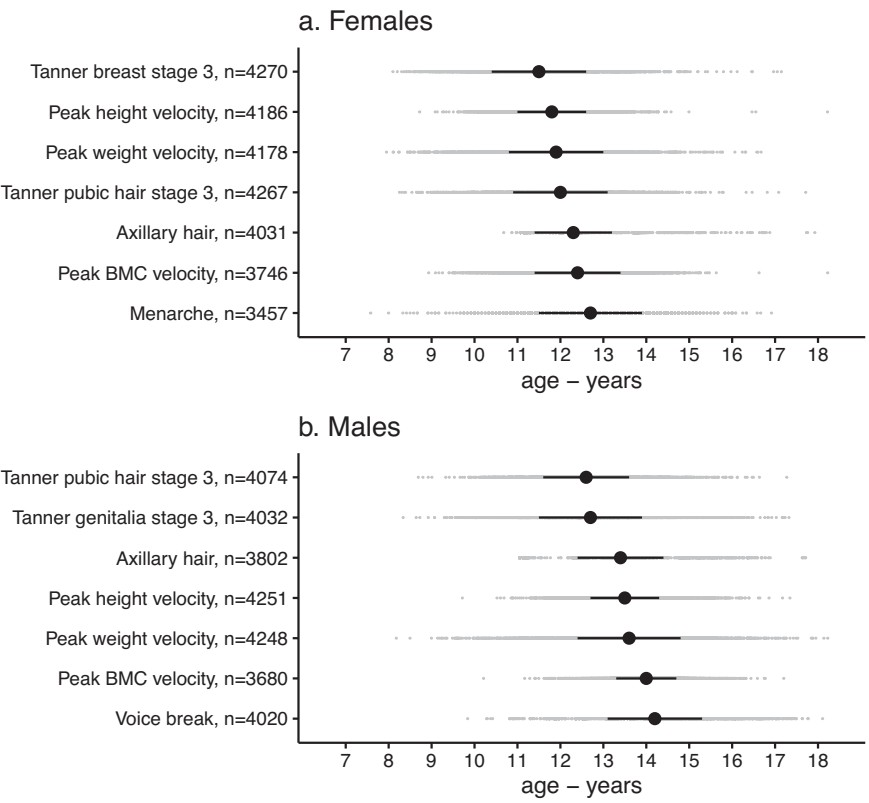

**Fig. 3 | Phenotypic correlations between indicator-based measures of pubertal age in females.** Figure presents density plots of each pubertal age measure, pairwise scatterplots and Pearson correlations between pubertal age measures. The sample size used for the correlation analysis was $n = 3037$ females that had complete data on all seven pubertal age measures. $P < 2 \times 10^{-16}$ for all correlations.

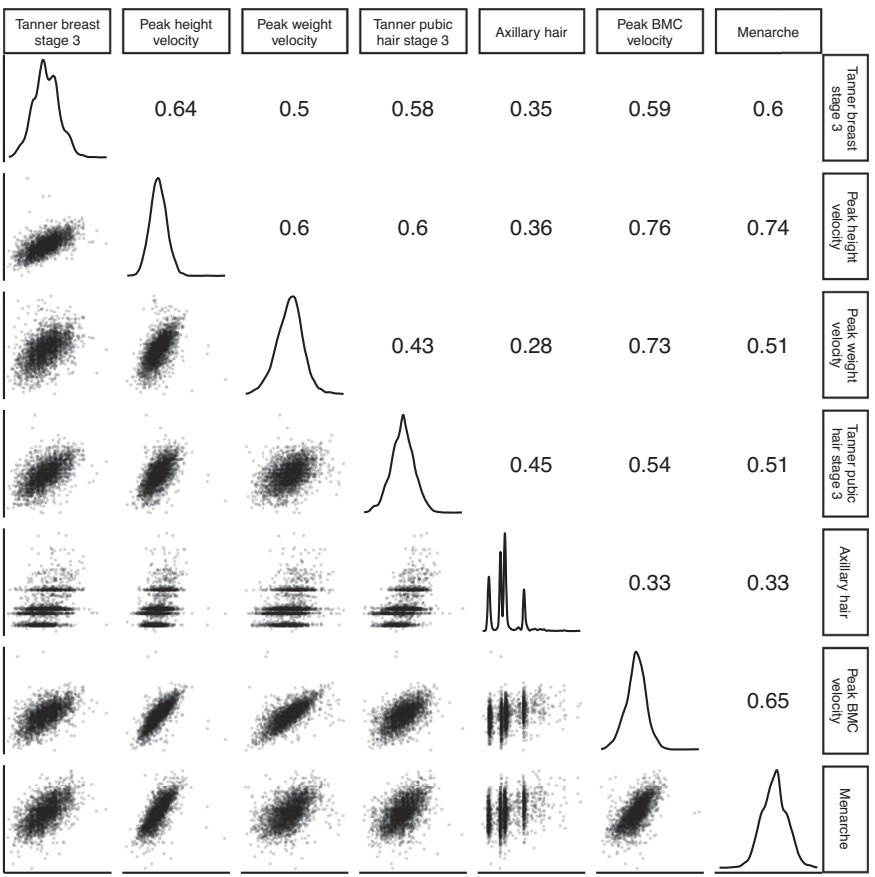

## Genetic correlations between indicator-based pubertal age measures

LDSR revealed mostly moderate to high genetic correlations between measures of pubertal age (Supplementary Data 1). This included genetic correlations between measures within each sex (for example, genetic correlation between age in Tanner pubic hair stage 3 and age at axillary hair in females was 0.87, $P = 0.002$), and between sex: both within measures (for example, genetic correlation between females and males for age at peak height velocity was 0.64, $P = 0.05$) and across different measures (for example, genetic correlation between age at menarche in females and age at peak BMC velocity in males was 0.78, $P = 0.007$).

## Associations of GRS's with indicator-based pubertal age measures

Higher GRSs, which were associated with older ages of female and male puberty, were both associated with older age of all derived pubertal age measures, and higher GRSs, which were associated with higher adulthood and childhood BMI, were both associated with younger age of all derived pubertal age measures, except for age in Tanner genitalia stage 3 (Fig. 5). The associations of pubertal timing GRSs with pubertal age measures were generally stronger for the female puberty timing GRS in females and were similar in magnitude for both scores in males. Associations of adulthood and childhood BMI GRSs were similar in magnitude for both scores in both females and males (Fig. 5).

## Association of childhood body composition with indicator-based pubertal age measures

Higher childhood fat mass index and lean mass index were both associated with younger age of puberty measures in females and males. The only exception was for age in Tanner genitalia stage 3 in males, where higher fat mass index was associated with older age (Fig. 6). Associations of fat mass and lean mass with measures of pubertal age were mostly similar in

magnitude in females and were stronger for fat mass in males. Association with younger age at peak weight velocity were noticeably stronger for fat mass than lean mass in both sexes (Fig. 6).

## Discussion

We used repeated assessments from a population-based cohort to examine and compare nine growth and development-based measures of pubertal timing. We found that, on average, breast development, appearance of pubic hair, and genitalia development were relatively early indicators of pubertal stage, while peak bone accrual, menarche, and voice breaking were later indicators. However, there was considerable variability between individuals in the timing of pubertal indicators. All pubertal age measures were inter-related, as demonstrated by positive phenotypic and genetic correlations. GRSs from large-scale GWAS's on the ages at menarche and voice breaking were positively associated with all other pubertal age measures, and GRS's for adulthood and childhood BMI were inversely associated with the pubertal age measures. Pre-pubertal fat mass and lean mass were inversely associated with all pubertal age measures, the only exception was a positive association between fat mass and genitalia stage in males.

To the best of our knowledge, ours is the first study to examine this collection of pubertal age measures. Our pubertal age estimates are consistent with studies that examined some of these measures. These include a study from the Danish National Birth Cohort (DNBC) on 14,000 participants with repeated data on the six developmental (but no anthropometric) measures which found that breast, genitalia, and pubic hair stages were early indicators of pubertal stage, with menarche and voice breaking being late indicators[35]. Our results agree with findings from the Edinburgh Long-itudinal Growth Study (ELGS) where height, menarche, and clinical examinations of development stages were taken every half-year until 20 years in 74 females and 103 males[31], and with a cross-sectional study of 703 Norwegian females aged 6-16 years that showed mean age in Tanner Stage 3 of breast and pubic hair development was younger than menarche[36]. Also

**Fig. 4 | Phenotypic correlations between indicator-based measures of pubertal age in males.** Figure presents density plots of each pubertal age measure, pairwise scatterplots and Pearson correlations between pubertal age measures. The sample size used for the correlation analysis was $n = 3139$ males that had complete data on all seven pubertal age measures. $P < 2 \times 10^{-16}$ for all correlations.

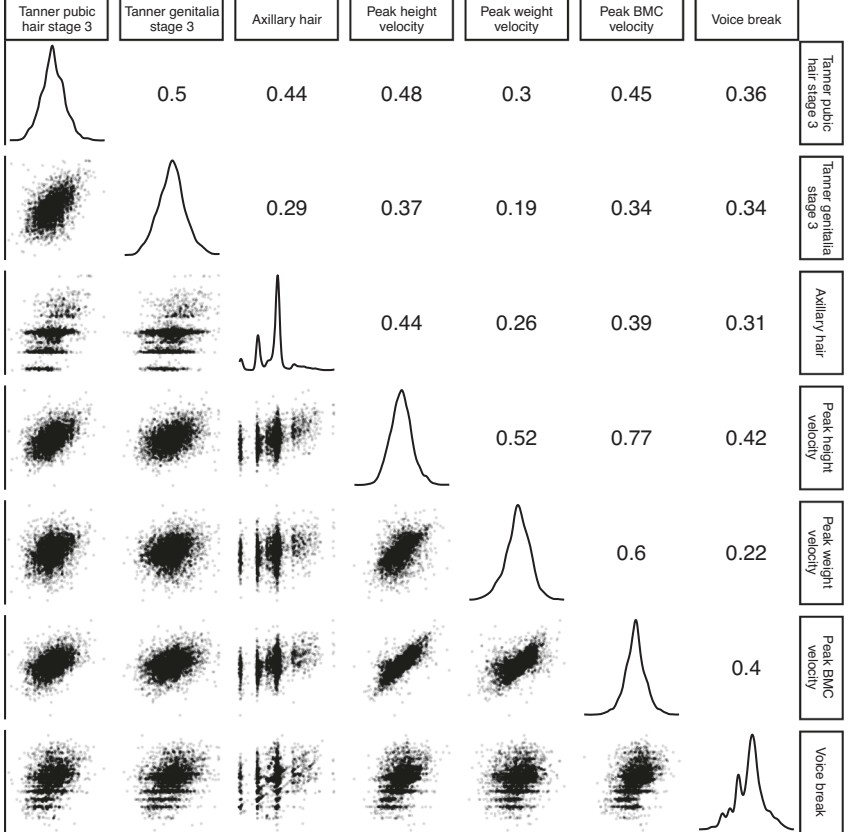

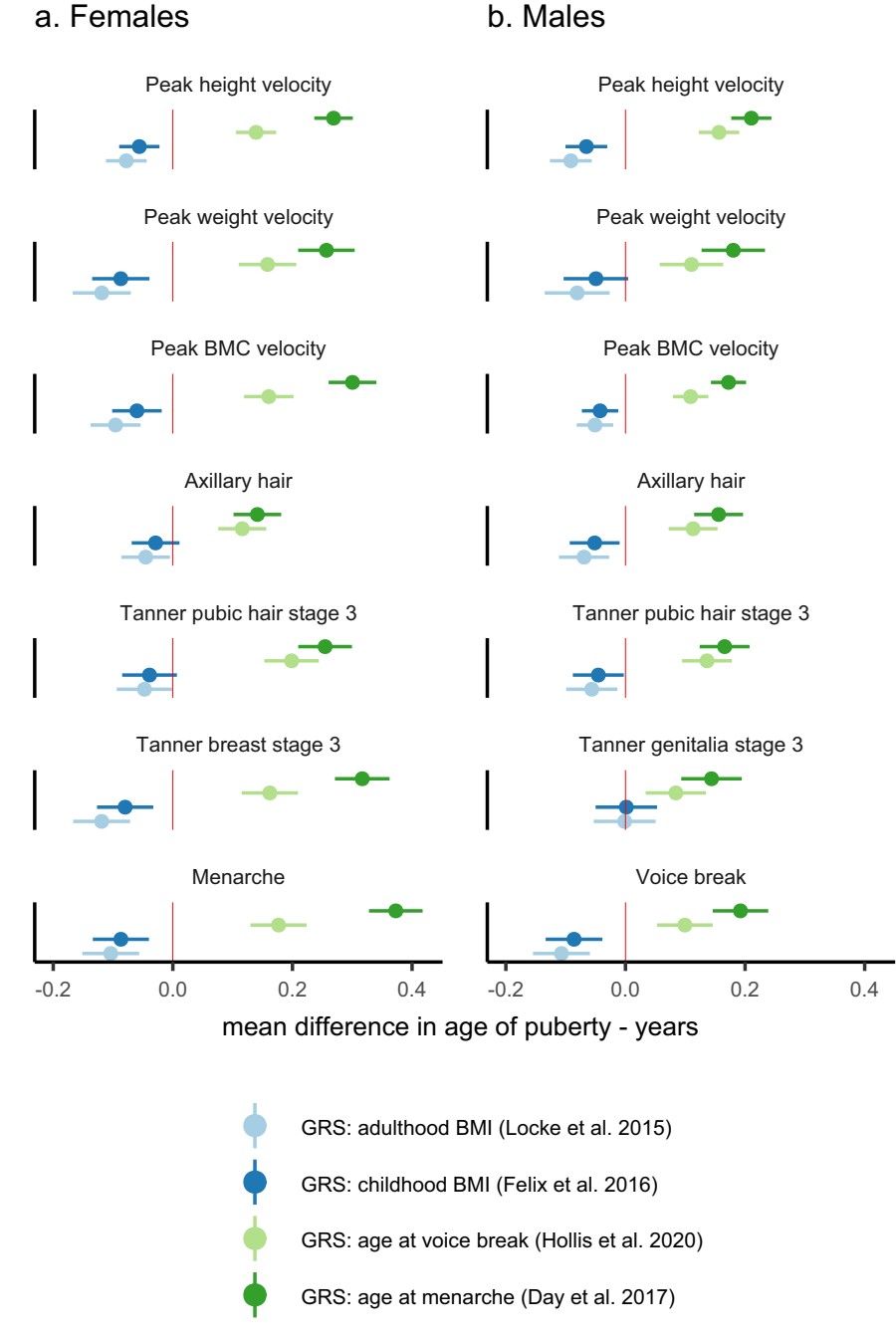

**Fig. 5 | Association between genetic risk scores and indicator-based measures of pubertal age.** Figure shows mean difference in age of each pubertal age measure per standard deviation increase in genetic risk score (GRS) for age at menarche, age at voice break, adulthood body mass index (BMI), and childhood BMI in females (**a**) and males (**b**). Points are mean differences and horizontal bars represent 95% confidence intervals. Estimates were obtained from separate unadjusted linear regression models for each pubertal age measure regressed on each GRS. The sample size for the analysis was $n = 2320$ females and $n = 2415$ males that had complete data on all four GRS and all pubertal age measures.

consistent with our estimates are findings from study of 105 twin pairs showing that mean age of peak velocity for height was slightly younger than for weight[37], and evidence from the US Bone Mineral Density in Childhood Study (BMDCS) that peak velocity occurred earlier for height than BMC[38]. Our observation of considerable variability in pubertal age between individuals, across all nine measures, is consistent with previous literature[4,5].

Our findings of positive phenotypic and genetic correlations between the pubertal age indicators are also consistent with studies that included some of these measures. For example, positive phenotypic correlations were found between measures in ELGS (r: 0.62 to 0.82 in males and r: 0.80 to 0.92 in females)[31], between voice breaking, axillary hair, and pubertal stages (r: 0.40 to 0.62) in a study of 730 Danish males[39], and between age of peak height and BMC velocity in BMDCS[38]. Like our LDSR results, Hollis et al.[9] reported a moderate genome-wide genetic correlation between age at voice breaking and age at menarche. Also in line with our findings are reports of

moderate to high genetic correlations (but with wide 95% CIs) between menarche and Tanner breast and pubic hair stage in 184 twin pairs[40], and between Tanner breast and pubic hair stage, and genitalia and pubic hair stage in 112 twin pairs[41]. Our study improves on these by including a larger sample size and examining genetic correlations across more measures.

We found that GRS's for childhood and adulthood BMI were both inversely associated with puberty timing measures, which is consistent with Mendelian randomization studies on age at menarche[12,13,42] and voice breaking[39]. Our finding of inverse associations between childhood fat mass and puberty timing is consistent with previous observations[39,43,44]. Our study adds to previous studies by comparing associations across nine measures of pubertal timing, showing that this association is substantially stronger for peak weight velocity than for other pubertal measures, and that childhood lean mass index is also inversely associated with the timing of pubertal indicators.

**Fig. 6 | Association of childhood fat mass and lean mass indices with indicator-based measures of pubertal age.** Figure shows mean difference in age of each pubertal measure per standard deviation higher pre-pubertal fat mass index and lean mass index (measured at mean age 10 years) in females (**a**) and males (**b**). Points are mean differences and horizontal bars represent 95% confidence intervals. Estimates were obtained from separate multivariable linear regression models for each pubertal age measure regressed on fat mass or lean mass, with adjustment for age at fat/lean mass assessment, childhood dietary intake, and maternal age at birth, maternal body mass index, maternal education, maternal smoking, and parity. The sample size for the analysis was $n = 1762$ females and $n = 2099$ males that had complete data on fat mass, lean mass, age at fat/lean mass assessment, all confounders, and all pubertal age measures.

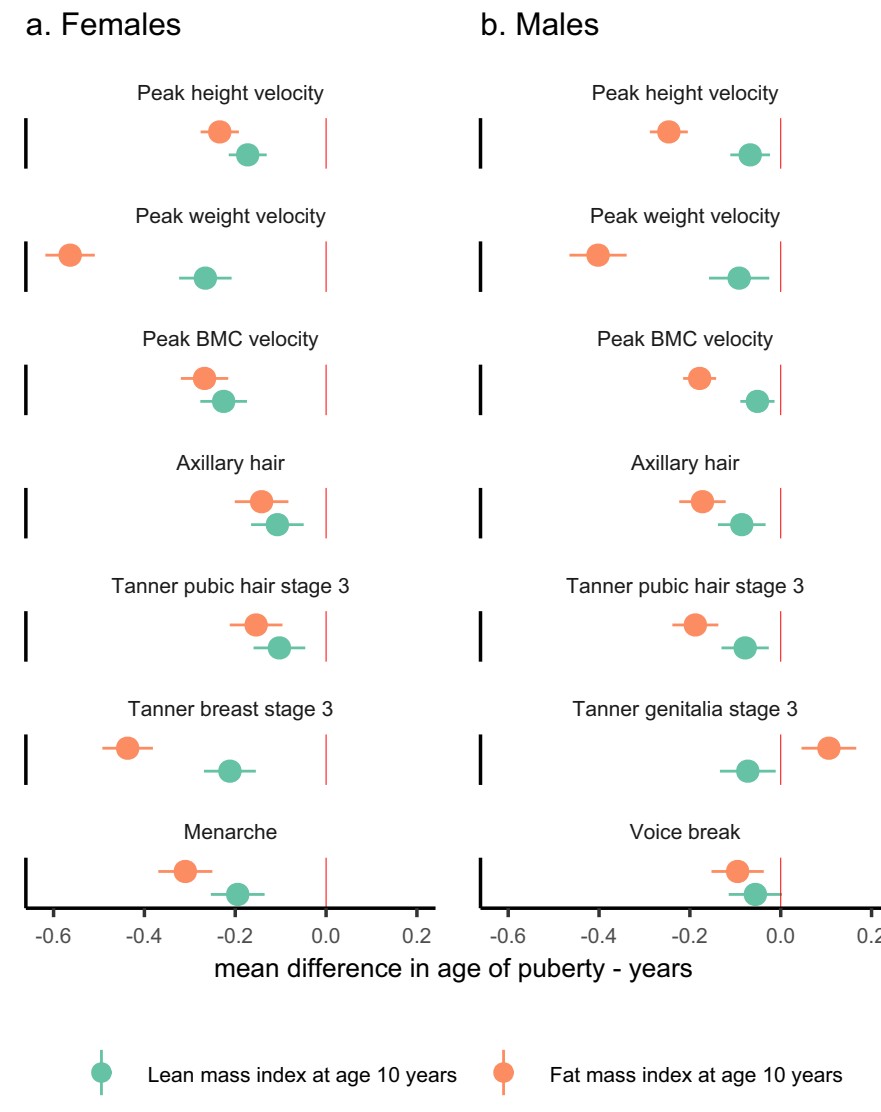

The age sequence of the different puberty measures is broadly consistent with the underlying molecular and hormonal changes driving appearance of these changes[2,3,45]. The substantial variability in pubertal timing between individuals may reflect between-individual differences in complex genetic and environmental factors (including exposures from early life onwards) contributing to puberty[5]. The positive phenotypic and genetic correlations between pubertal age measures suggest that they all might capture the same process and have a shared heritable contribution (from common genetic variation)[6]. Our finding of positive genetic correlations between males and females, which were generally lower than those within sex, point to both similar genetic factors driving pubertal timing in each sex as well sex-specific genetic effects on pubertal timing[6,46].

GRSs from published genome-wide significant SNPs for age at menarche and voice breaking associated positively with all other pubertal age measures. While replication in independent cohorts is needed, if confirmed, this (and the positive phenotypic and genetic correlations between measures) suggests our suite of nine measures could all be used as measures of pubertal age when assessing the determinants and effects of pubertal timing, and can facilitate research in cohorts with repeated assessments, e.g., to assess whether associations with risk factors or outcomes are comparable across all measures or if they are specific to certain growth/development measures (and sex).

We found that GRS's for childhood and adulthood BMI were inversely associated with most pubertal age measures, which suggests that children with higher adiposity are more likely to experience earlier puberty[42], possibly through adiposity-related hormonal perturbations[47] and those with earlier puberty may be more likely to have higher adiposity in adulthood, possibly due to shared genetic contributions to childhood adiposity[13,48]. Our finding that both higher childhood fat mass and lean mass were associated with earlier puberty supports a role for higher childhood body size beyond solely adiposity in earlier pubertal timing. In contrast to the other pubertal indicators, higher childhood fat mass was associated with older (rather than younger) Tanner genitalia stage, and childhood BMI GRS was not associated with genitalia stage, which could both be due to Tanner staging being more challenging to implement in overweight or obese children[20], or because the more adipose children were more likely to exaggerate their development[49].

Data on developmental measures were collected by questionnaire using parent/self-reporting which might result in larger measurement errors compared with growth measures, and these differences in measurement error might have biased observed differences in pubertal ages[17]. Assessment of Tanner stages was supported by pictorial depictions and accompanying explanations of each Tanner stage which might have mitigated against this[49]. Furthermore, studies that have used clinical assessments (i.e., observation by trained clinicians or research staff) rather than self-report have reported

similar results to ours[31,39]. Axillary hair was collected as a dichotomous response, which could result in an imprecise estimate of pubertal age. Only five repeated measures of BMC were available for deriving the age of peak BMC velocity which may have led to imprecise estimation[50]. GWAS sample sizes were small in ALSPAC which can lead to unstable LDSR genetic correlation estimates. While analyses of pre-pubertal body composition were adjusted for measured confounders, we cannot rule out bias from residual or unmeasured confounding. ALSPAC participants were White Europeans and results might not generalise to other ethnic groups. Other pubertal age measures such as age of first ejaculation, and skeletal bone age were not available, and could have provided further information on pubertal timing.

In summary, findings from this prospective population-based cohort study of males and females supported all nine growth and development-based pubertal age measures as consistent measures of age at puberty, by providing evidence that they are measuring the same biological process. Choice of measure(s) to use in studies with plans for data collection is influenced by various factors, including research questions, available resources together with competing demands for other types of data to be collected, participant burden, and acceptability of data collection methods. For instance, studies comparing pubertal timing between males and females could focus on the measures available in both sexes, such as height, weight, and BMC, as well as pubic or axillary hair. Collecting longitudinal growth and development data can be challenging due to limited funding and research resources. Cohort studies that collect repeated data prospectively are research resources, often available to the global research community rather than funded to address a limited set of research questions. Thus, repeated height or weight data collections, which are likely to be relevant to many areas of study might become the basis for assessing pubertal age. However, there would be scientific value in other studies measuring as many of the measures we present so our finding might be replicated in independent studies. Further, availability of multiple measures would allow the comparison of risk factors and outcome across pubertal measures. Finally, the correlations presented may be useful for harmonising measures across studies (e.g., meta-analysis).

## Data availability

Researchers interested in accessing ALSPAC data used in this study will need to submit a research proposal (https://proposals.epi.bristol.ac.uk/) for consideration by the ALSPAC Executive Committee (managed access). The ALSPAC Executive Committee encourage and facilitate data sharing with all 'bona fide' researchers. A bona fide researcher is defined as being a person with professional expertise to conduct bona fide research; and who has a formal affiliation with a bona fide research organisation that requires compliance with appropriate research governance and management systems. The ALSPAC data are not publicly available because the Executive Committee needs to check that the applicant is a bone fide researcher and that the proposed research is in the public interest. Source data underlying the graphs and charts presented in Figs. 1–6 can be found in Supplementary Data 2. All other data are available from the corresponding author on reasonable request.

## Code availability

Statistical code (and analysis plan) used for this paper can be found in the Open Science Framework website at https://osf.io/3qndg/[25].

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

## Acknowledgements

We are extremely grateful to all the families who took part in this study, the midwives for their help in recruiting them and the whole ALSPAC team, which includes interviewers, computer and laboratory technicians, clerical workers, research scientists, volunteers, managers, receptionists, and nurses. ALSPAC data were collected and managed using REDCap (Research Electronic Data Capture) electronic data capture tools hosted at the University of Bristol. GWAS data was generated by Sample Logistics and Genotyping Facilities at Wellcome Sanger Institute and LabCorp (Laboratory Corporation of America) using support from 23andMe. This project has received funding from the European Union's Horizon 2020 research and innovation programme under grant agreement No. 874739 (LongITools). A.E. and D.A.L. receive part of their salary from the European Union's Horizon 2020 research and innovation programme under grant agreement No. 101021566 (ART-HEALTH). A.E., M.F., A.G.S., J.A.B., J.H., L.D.H., K.T., N.J.T. and D.A.L. work in a Unit that receives funds from the University of Bristol and UK Medical Research Council (MC_UU_00032/05 and MC_UU_00032/02). D.A.L. is a National Institute of Health Research Senior Investigator (NF-0616-10102) and is also supported by a British Hear Foundation Chair (CH/F/20/90003). The UK Medical Research Council and Wellcome (Grant ref: 217065/Z/19/Z), and the University of Bristol provide core support for ALSPAC. A comprehensive list of grants funding is available on the ALSPAC website (http://www.bristol.ac.uk/alspac/external/documents/grant-acknowledgements.pdf). The funders had no role in the design and conduct of the study; collection, management, analysis, and interpretation of the data; preparation, review, or approval of the manuscript; and decision to submit the manuscript for publication. A.E. had full access to all the data in the study and takes responsibility for the integrity of the data and accuracy of the data analysis.

## Author contributions

A.E. developed the idea for the paper with initial input from D.A.L. and further input from all authors. A.E. developed the analysis plan with input from all authors. A.E. did the majority of the statistical analysis. M.F. did the LDSR genetic correlation analysis. A.G.S. calculated the GRS for age at voice break. J.A.B. calculated the GRS for age at menarche, adulthood BMI, and childhood BMI. T.J.C., J.H. and K.T. provided advice on fitting mixed effects models to estimate age of puberty. A.E. wrote the first draft of the manuscript. M.F., A.G.S., J.A.B., T.J.C., J.H., L.D.H., S.S., K.T., N.J.T. and D.A.L. provided feedback on the draft and approved the final version for submission.

## Competing interests

The authors declare the following competing interests: D.A.L. reported grants from national and international government and charity funders, Roche Diagnostics, and Medtronic Ltd for work unrelated to this publication. The other authors declare no competing interests.
