## [Peer Review File · Communications Medicine]

This manuscript has been previously reviewed at another Nature Portfolio journal. This document only contains reviewer comments and rebuttal letters for versions considered at *Communications Medicine*.Reviewers' comments:

Reviewer #1 (Remarks to the Author):

Understanding the timing of growth events and milestones is important when assessing childhood health. Accelerated or delayed onset of puberty can have long-term effects if not identified. As stated in the title the manuscript seeks to compare nine growth and development measures of pubertal timing. This is timely and important topic. The sample is outstanding including and includes longitudinal observations of up to 8,500 individuals. Unfortunately, the work suffers from poor understanding of previous work, vague presentation, and confusing analyses.

Understanding of previous work.

There are statements throughout that declare little work has been done on pubertal timing. These include: "Most research relating to puberty timing has relied on reported age at menarche (a notable singular event, with no clear male equivalent) and therefore has been undertaken in females only." Or "to the best of our knowledge, no study has systematically compared anthropometric and developmental measures of pubertal age" (both P.3, Paragraph 2). These comments are particularly surprising/frustrating as 1) the references cited in this paper identify multiple papers on male puberty and growth and puberty and 2) a simple PubMed search identifies hundreds of papers that cover these topics.

Vague presentation.

The manuscript uses nine indicators of puberty (e.g., pubic hair development, breast development, etc.) along with anthropometric measures of growth (height, weight, BMC). The use the term Pubertal age to refer to both timing of pubertal indicators and overall timing of Puberty. As puberty is a period of change spanning multiple years the term "Pubertal age" is confusing. There is no mention, in the introduction, that demographic factors such as maternal pregnancy, maternal education, parity, etc. were collected or how they fit into the general plan of the analysis. The results section shows these were used as confounders in the models of body composition, but it is not clear if they were included in other models.

Confusing analysis.

The text states: "Except for age at menarche, which was calculated as the first reported age, all pubertal age measures were derived by applying mixed effects models to each set of repeated puberty assessments (Fig 1) and identifying the age corresponding to the peak of the velocity curve." (Page 5, paragraph 1). Figure 1, however, does not show figures associated with a mixed effect model, rather it displays stacked histograms for pubertal indicators and a line plot for anthropometric measures (with SD at each visit).

Page 5, paragraph 2 describes "Mean age of puberty" (actually indicator-based age) as "11.5 years for age in Tanner breast stage" etc. The ages given here are actually based on the information in Figure 2. In this figure, two indicator-based pubertal ages are presented (the median and the mean). The mean is the value used in the text. The values in these cases do not appear to be based on mixed effects models as described in the text and, while the figure and table presented in Figure 2 show the tremendous age range associated with attainment of these indicators, there is not an adequate discussion of this. This is extremely important, it is highly misleading to treat a simple measure of central tendency such as mean or median as truth. So, while Tanner breast stage 3 may have a median ~11.5, the range of values is 10 years. The discussion should focus much more on the variation in timing than on trying to craft a story out of a single number.

Phenotypic correlations.

The staging used in this paper for pubertal indicators (Tanner Staging) was created using an age-graded sample and so it is not surprising that there is a modest correlation between stages in different indicators. Ignoring this, it is noted that p values are not provided for correlation values nor is there any adjustment for multiple testing.

SITAR models. Sitar is a powerful method for modelling longitudinal growth. The description of its use in the current paper is poorly described. Results are only provided in the Supplementary material.

Conclusion

I have tried to touch upon some of the main issues with the current manuscript. Overall, there is no clear hypothesis being tested here, and, if this paper is designed as a descriptive analysis of pubertal staging and age there are numerous examples in existence.

Reviewer #2 (Remarks to the Author):

In this manuscript, Elhakeem et al. provide a systematic analysis of different indicators of the timing of puberty. The analysis is thorough, well conducted, well written, and will be of high interest for researchers of the field. I found the addition of genetic analyses particularly interesting. I strongly support the publication of this work and mainly have questions that were triggered by the data presented - if the authors feel that other readers would encounter them it would be much appreciated to see the text extended to provide some answers. All my congratulations for this well executed and very interesting study.

Main comments

- Strong outliers for growth velocity measures: Both box plots and correlation plots show strong outliers and samples decorrelating for growth velocity measures. I was wondering whether these can be due to a low number of anthropometric values for these samples that make it challenging for SITAR to correctly track the growth velocity? I did not find any exclusion criteria based on the number of values available for the growth curves. While I acknowledge that few outliers will not affect the overall conclusions of the authors, it would be good to investigate these outliers and make sure that enough data points are available to correctly model the growth.

- In their comparison of the timing of the different measures, how consistent is the order? Do kids follow the stages of puberty timing in more or less the same order or are these unrelated?

- Interpretation on the low LDSC: can the authors extend in terms of possible genetic underpinning and complexity of puberty?

- Childhood BMI: Locke et al. contains mainly older kids for which puberty might have begun. Can this influence the conclusions of the authors, and notably alter the comparison between boys and girls?

- Lack of agreement between childhood BMI and Tanner: can it be due to the fact that the tanner scale is more challenging to implement on children with high BMI?

Minor text edits

> Most research relating to puberty timing has relied on reported age at menarche (a notable singular event, with no clear male equivalent) and therefore has been undertaken in females only. This seems to contradict the rest of the text and feels unspecific/wrong. Please consider rephrasing or removing this statement.

> page 5 "age in age in Tanner pubic hair"
Please correct the repetition of "age in"

> this suit of nine measures
Suite?

> In the discussion on the partial agreement between measures, I recommend also including <https://doi.org/10.1210/clinem/dgaa107>

RE: COMMSMED-23-0482-T Evaluation and comparison of nine growth- and development-based measures of pubertal timing

We thank the editor for considering our manuscript and the reviewers for their helpful comments, which have improved our manuscript. Please see below for our point-by-point response to reviewer comments, and details of changes made, which are all highlighted using red text in the revised manuscript.

REVIEWER #1:

1. Understanding the timing of growth events and milestones is important when assessing childhood health. Accelerated or delayed onset of puberty can have long-term effects if not identified. As stated in the title the manuscript seeks to compare nine growth and development measures of pubertal timing. This is timely and important topic. The sample is outstanding including and includes longitudinal observations of up to 8,500 individuals. Unfortunately, the work suffers from poor understanding of previous work, vague presentation, and confusing analyses.

Understanding of previous work.

There are statements throughout that declare little work has been done on pubertal timing. These include: "Most research relating to puberty timing has relied on reported age at menarche (a notable singular event, with no clear male equivalent) and therefore has been undertaken in females only." Or "to the best of our knowledge, no study has systematically compared anthropometric and developmental measures of pubertal age" (both P.3, Paragraph 2). These comments are particularly surprising/frustrating as 1) the references cited in this paper identify multiple papers on male puberty and growth and puberty and 2) a simple PubMed search identifies hundreds of papers that cover these topics.

Response: We have edited the above paragraph in the introduction to remove the highlighted statements and better acknowledge previous work.

Page 4:

There is substantial variation in the age of puberty between children^{4,5}, which is attributable to genetic as well as non-genetic factors, such as nutrition⁶⁻⁹. Understanding determinants of variation in puberty timing between individuals is important given its relation to reproductive capability and social and health implications, including the risk of some cancers⁸⁻¹⁶. Within an individual, there is variation in the timing of different maturation processes (e.g., skeletal, and sexual maturation), and in related structures within a maturational process (e.g., within sexual maturation, pubic hair and genitalia can have different levels of maturity)¹⁷. Various approaches and indicators have been used by studies to measure puberty timing¹⁷⁻²¹, which have included self/parent-reported age at menarche and voice change in females and males, respectively, and longitudinally modelled age at peak height velocity in both. As no single measure can capture all maturational processes during puberty, detailed, systematic analysis of several anthropometric and developmental measures of puberty timing, including in both sexes, can help reveal their value for life course research, and identify which measure is best for exploring

the causes and consequences of pubertal timing, and what might be done to mitigate those effects.

Vague presentation.

2. The manuscript uses nine indicators of puberty (e.g., pubic hair development, breast development, etc.) along with anthropometric measures of growth (height, weight, BMC). The use the term Pubertal age to refer to both timing of pubertal indicators and overall timing of Puberty. As puberty is a period of change spanning multiple years the term “Pubertal age” is confusing.

Response: Re have replaced ‘pubertal age’ with ‘timing of pubertal indicators’ and ‘indicator-based age of puberty’ throughout the manuscript.

3. There is no mention, in the introduction, that demographic factors such as maternal pregnancy, maternal education, parity, etc. were collected or how they fit into the general plan of the analysis. The results section shows these were used as confounders in the models of body composition, but it is not clear if they were included in other models.

Response: We have moved the Methods section up to before the Results section (which is in line with the requirements for Communications Medicine) and added a statement in the methods section to clarify why, and how, these were used in our study.

Page 9:

“Childhood body composition measurements and confounders

Pre-pubertal fat mass index (total body fat mass divided by height squared) and lean mass index (total body lean mass divided by height squared), both in units of kg/m², were derived from DXA scans performed at mean age 9.9 years and were used to examine associations of childhood body composition with puberty timing. DXA scans were performed using a Lunar Prodigy scanner (Lunar Radiation Corp) and were analysed according to the manufacturer’s standard scanning software and positioning protocols. Scans were reanalysed as necessary to ensure optimal placement of borders between adjacent subregions, and scans with anomalies were excluded. Exact age in months when scan was performed was recorded. Fat mass and lean mass indices were standardised (to mean=0 and SD=1) prior to examining association with the derived indicator-based pubertal ages.

Maternal education, maternal early pregnancy BMI and early pregnancy smoking, maternal age at birth, parity, and child’s diet were identified as factors that could plausibly influence both child body composition and puberty timing and selected to be included as confounder adjustment when examining associations of childhood fat mass and lean mass indices with puberty timing. Maternal confounders were reported using questionnaires during pregnancy (maternal BMI was calculated from reported height and weight). Child’s diet was based on daily energy intake (in kilojoules per day) and derived from food frequency questionnaires completed by the parent when the child was aged 7 years.”

Confusing analysis.

4. The text states: “Except for age at menarche, which was calculated as the first reported age, all pubertal age measures were derived by applying mixed effects models to each set of repeated puberty assessments (Fig 1) and identifying the age corresponding to the peak of the velocity curve.” (Page 5, paragraph 1). Figure 1, however, does not show figures associated with a mixed effect model, rather it displays stacked histograms for pubertal indicators and a line plot for anthropometric measures (with SD at each visit).

Response: Apologies for the confusion. We have edited the text to make it clear that Fig 1 summarises the observed (i.e., collected) data that were used to derive the 9 indicator-based pubertal age measures, and this figure is now cited in the Methods section instead of the results.

Page 6:

*“Data used to derive indicator-based pubertal ages were collected prospectively using nine repeated research clinic assessments and nine puberty-specific questionnaires. **Figure 1** summarises the observed data from these clinic assessments and questionnaires, and **Supplementary Table 1** and **Supplementary Table 2** provide more information.”*

5. Page 5, paragraph 2 describes “Mean age of puberty” (actually indicator-based age) as “11.5 years for age in Tanner breast stage” etc. The ages given here are actually based on the information in Figure 2. In this figure, two indicator-based pubertal ages are presented (the median and the mean). The mean is the value used in the text. The values in these cases do not appear to be based on mixed effects models as described in the text and, while the figure and table presented in Figure 2 show the tremendous age range associated with attainment of these indicators, there is not an adequate discussion of this. This is extremely important, it is highly misleading to treat a simple measure of central tendency such as mean or median as truth. So, while Tanner breast stage 3 may have a median ~11.5, the range of values is 10 years. The discussion should focus much more on the variation in timing than on trying to craft a story out of a single number.

Response: Sorry for the confusion. The ages presented in Figure 2 and the text are those derived from the mixed effects models, except for age at menarche (because that was not modelled). We have edited the text in the methods to clarify. We have added text to the results and discussion section on the variation in the timing of puberty indicators.

Page 10:

“Age at menarche was calculated as the first reported age at onset of menstruation. Pubertal ages for all other measures were derived using the SITAR (Super Imposition by Translation And Rotation) method of growth curve analysis²⁷.”

Page 14-15:

“There was considerable variability between individuals in the timing of pubertal indicators, e.g., in females, the SD around mean ages ranged from 0.8 (peak height velocity) to 1.2 years (menarche), and in males, from 0.7 years (peak BMC velocity) to 1.2 years (Tanner genitalia stage 3 and peak weight velocity) (Fig. 2). Moreover, age in Tanner breast stage 3 occurred first in 38.5% of females and age at menarche was last in 41.8% and likewise, age in Tanner pubic hair stage 3 occurred first in 36.4% of males and age at voice break was last in 45.8% of males. Removing outliers minimally influenced the estimated age at peak height, weight, and BMC velocity (number of observations removed in females and males were: 100/26869 and 895/25898 for height, 129/26765 and 269/25792 for weight, and 6/12854 and 42/11823 for BMC). Following removal of outliers, mean (and SD) ages at peak height, weight, and BMC velocity respectively were 11.5 (1.2), 11.9 (1.2), and 12.5 (1.0) years in females and 13.1 (1.1), 13.5 (1.3), and 13.2 (1.3) years in males.”

Page 17:

“We found that, on average, breast development, appearance of pubic hair, and genitalia development were relatively early indicators of pubertal stage, whilst peak bone accrual, menarche, and voice breaking were later indicators. However, there was considerable variability between individuals in the timing of pubertal indicators.”

Page 18:

“Our observation of considerable variability in pubertal age between individuals, across all nine measures, is consistent with previous literature^{4,5}.”

Page 19:

“The substantial variability in pubertal timing between individuals may reflect between-individual differences in complex genetic and environmental factors (including exposures from early life onwards) contributing to puberty⁵.”

6. Phenotypic correlations.

The staging used in this paper for pubertal indicators (Tanner Staging) was created using an age-graded sample and so it is not surprising that there is a modest correlation between stages in different indicators. Ignoring this, it is noted that p values are not provided for correlation values nor is there any adjustment for multiple testing.

Response: We are not sure what the reviewer means by age-graded sample. The reviewer is correct that we chose to omit P-values for the correlations, as even very modest correlations (e.g. $r = 0.1$) are highly significant given the large sample size (and without P-values, no testing is being done). We have nevertheless added a footnote to Figure 3 and Figure 4 stating that P was $<2 \times 10^{-16}$ for all pairwise correlations.

7. SITAR models. Sitar is a powerful method for modelling longitudinal growth. The description of its use in the current paper is poorly described. Results are only provided in the Supplementary material.

Response: We have edited the text in the Methods to improve the description of the SITAR models.

Page 10-12:

“Derivation of indicator-based pubertal age measures

Nine indicator-based puberty timing (age) measures were derived: two in females only (age at menarche and age in Tanner breast stage 3), two in males only (age at voice breaking and age in Tanner genitalia stage 3), and five in both females and males (age at peak BMC, height, and weight velocity, age in Tanner pubic hair stage 3, and age at axillary hair).

Estimated pubertal ages were analysed in months for all measures and presented in years to aid interpretation. All analyses were restricted to White ethnicity individuals (>95% of all participants) to enable consistency across phenotypic and genetic analyses. Analyses were performed in R version 4.02 (R Project for Statistical Computing).

Age at menarche was calculated as the first reported age at onset of menstruation. Pubertal ages for all other measures were derived using the SITAR (Super Imposition by Translation And Rotation) method of growth curve analysis²⁷. SITAR is a shape invariant nonlinear mixed effects model that fits a single (mean) natural spline growth curve in the study sample and tailors it (using random effects) to define how individual growth curves differ from the mean curve. The SITAR model usually has up to three random effects that describe the size, timing, and intensity of individual growth relative to the mean growth curve. Size adjusts for differences in growth and geometrically reflects up or down shifts in the mean curve, timing adjusts for differences in the timing of peak growth and geometrically reflects left to right shifts in the mean curve, and intensity adjusts for the duration of the growth spurt and geometrically corresponds to shrinking or stretching of the age scale (which rotates the mean curve)²⁷. A recent addition to the SITAR software allows a fourth ‘post-growth’ random effect to be fitted which extends SITAR to model variability in the adult slope of the growth curve to allow post-pubertal growth rate to vary between individuals²⁸.

Height was modelled using the standard SITAR approach with three random effects. Weight and BMC were modelled using SITAR with all four random effects to allow for variation in growth post-puberty. Tanner stages for pubic hair, breast, and genitalia development, and voice breaking, and axillary hair were modelled using SITAR with up to two random effects for timing and intensity²⁹. This reduced SITAR model (i.e., without the size random effect) was used as all individuals are measured on the same 5-point scale (or 3 for voice breaking, and 2 for axillary hair), so their position on the scale at any particular time depends purely on their developmental age at that time, taking into account their timing and intensity effects.

SITAR models were fitted separately in males and females with at least one outcome measurement. The best fitting models were identified by comparing models with 2 to 5 knots (placed at quantiles of the age distribution) in the mean spline curve and inspecting the fitted mean curves and the Bayesian information criterion (BIC) values

for each model (**Supplementary Table 4, Supplementary Figures 3-4**). Covariances for the random effects were modelled (**Supplementary Table 5**). Indicator-based pubertal age was estimated using the timing random effect from each SITAR model and represent age at peak growth velocity for height, weight and BMC, age in Tanner stage 3 of pubic hair, breast (females only) and genitalia (males only) development, and age at voice breaking (males only) and axillary hair appearance. Because regression modelling can allow for measurement error, inconsistent responses (i.e., reporting a developmental stage that was lower than that reported in a previous questionnaire) were included in the analysis, except for inconsistent responses in voice breaking which were removed prior to modelling due to convergence issues. To examine the effect of outliers on anthropometric pubertal age estimates, we refitted models for height, weight, and BMC and re-estimated ages at peak velocity after removing conventional putative outliers (+/- 5 SD).”

8. Conclusion

I have tried to touch upon some of the main issues with the current manuscript. Overall, there is no clear hypothesis being tested here, and, if this paper is designed as a descriptive analysis of pubertal staging and age there are numerous examples in existence.

Response: Thank you for the above points which have improved the paper. The aim of our study was to quantitatively evaluate and compare growth and development-based measures of pubertal timing. Our hypothesis is that these comparisons could help better characterisation of variation in pubertal timing and hence determining its causes and outcomes of this variation across the life course in future research. While there are examples of similar work, we are not aware of any other studies that have simultaneously examined this collection of indicators, and in this large a sample size, or that have concurrently examined phenotypic and genetic correlations.

REVIEWER #2:

9. In this manuscript, Elhakeem et al. provide a systematic analysis of different indicators of the timing of puberty. The analysis is thorough, well conducted, well written, and will be of high interest for researchers of the field. I found the addition of genetic analyses particularly interesting. I strongly support the publication of this work and mainly have questions that were triggered by the data presented - if the authors feel that other readers would encounter them it would be much appreciated to see the text extended to provide some answers. All my congratulations for this well executed and very interesting study.

Response: Thank you for the positive assessment.

Main comments

10. Strong outliers for growth velocity measures: Both box plots and correlation plots show strong outliers and samples decorrelating for growth velocity measures. I was wondering whether these can be due to a low number of anthropometric values for these samples that make it challenging for SITAR to correctly track the growth velocity? I did not find any exclusion criteria based on the number of values available for the growth curves. While I acknowledge that few outliers will not affect the overall conclusions of the authors, it would be good to investigate

these outliers and make sure that enough data points are available to correctly model the growth.

Response: As the reviewer mentions, few outliers are unlikely to influence our findings and we feel that the benefit of including all observations outweighs the benefit of exclusion of 'outliers' due to loss of data. However, as recommended, we have added a sensitivity analysis that removes outliers and reruns models for the three anthropometric measures (this did not alter any of the research conclusions). We have edited the manuscript to report this analysis.

Page 12:

"Lastly, we did a sensitivity analysis to examine the effect of outliers on anthropometric pubertal age estimates by refitting SITAR models for height, weight, and BMC and re-estimating ages at peak velocity after removing conventional putative outliers (+/- 5 SD)."

Page 15:

"Removing outliers had minimal impact on the estimated age at peak height, weight, and BMC velocity (the number (%) of observations removed in females and males were 99 (0.4%) and 633 (2.4%) for height, 119 (0.4%) and 238 (0.9%) for weight, and 6 (0.0004%) and 42 (0.4%) for BMC. Following removal of outliers, mean (and SD) ages at peak height, weight, and BMC velocity respectively were 11.5 (1.2), 11.8 (1.2), and 12.5 (1.0) years in females and 13.4 (1.1), 13.8 (1.3), and 13.2 (1.3) years in males."

11. In their comparison of the timing of the different measures, how consistent is the order? Do kids follow the stages of puberty timing in more or less the same order or are these unrelated?

Response: As highlighted by Reviewer 1, there was considerable variation in the age of puberty for each indicator. We have edited the text to expand on this variation and to note the limitation of focusing on average or mean age (see our response to #5).

12. Interpretation on the low LDSC: can the authors extend in terms of possible genetic underpinning and complexity of puberty?

Response: We have added statements to the discussion on the LDSC results, in terms of its relation to previous studies and interpretation.

Page 17-18:

"To the best of our knowledge, ours is the first study to examine this collection of pubertal age measures. Our pubertal age estimates are consistent with studies that examined some of these measures. These include a study from the Danish National Birth Cohort (DNBC) on 14,000 participants with repeated data on the six developmental (but no anthropometric) measures which found that breast, genitalia, and pubic hair stages were early indicators of pubertal stage, with menarche and voice breaking being late indicators³⁴. Our results agree with findings from the

Edinburgh Longitudinal Growth Study (ELGS) where height, menarche, and clinical examinations of development stages were taken every half-year until 20 years in 74 females and 103 males²⁹, and with a cross-sectional study of 703 Norwegian females aged 6-16 years that showed mean age in Tanner Stage 3 of breast and pubic hair development was younger than menarche³⁵. Also consistent with our estimates are findings from study of 105 twin pairs showing that mean age of peak velocity for height was slightly younger than for weight³⁶, and evidence from the US Bone Mineral Density in Childhood Study (BMDCS) that peak velocity occurred earlier for height than BMC³⁷. Our observation of considerable variability in pubertal age between individuals, across all nine measures, is consistent with previous literature^{4,5}.

Our findings of positive phenotypic and genetic correlations between the pubertal indicators are also consistent with studies that included some of these measures. For example, positive phenotypic correlations were found between measures in ELGS (r : 0.62 to 0.82 in males and r : 0.80 to 0.92 in females)²⁹, between voice breaking, axillary hair, and pubertal stages (r : 0.40 to 0.62) in a study of 730 Danish males³⁸, and between age of peak height and BMC velocity in BMDCS³⁷. Like our LDSR results, Hollis et al⁹ reported a moderate genome-wide genetic correlation between age at voice breaking and age at menarche. Also in line with our findings are reports of moderate to high genetic correlations (but with wide 95% CIs) between menarche and Tanner breast and pubic hair stage in 184 twin pairs³⁹, and between Tanner breast and pubic hair stage, and genitalia and pubic hair stage in 112 twin pairs⁴⁰. Our study improves on these by including a larger sample size and examining genetic correlations across more measures.”

Page 19:

“The positive phenotypic and genetic correlations between pubertal age measures suggest that they all might capture the same process and have a shared heritable contribution (from common genetic variation)⁶. Our finding of positive genetic correlations between males and females, which were generally lower than those within sex, point to both similar genetic factors driving pubertal timing in each sex as well sex-specific genetic effects on pubertal timing^{6,45}.”

13. Childhood BMI: Locke et al. contains mainly older kids for which puberty might have begun. Can this influence the conclusions of the authors, and notably alter the comparison between boys and girls?

Response: We used Locke et al to derive our *adulthood* (not childhood) BMI GRS. Our childhood BMI GRS was derived from *Felix et al* which only included children aged 3-10 years. This has been made clearer in the manuscript.

Page 8-9:

“GRSs for female and male puberty timing, and adulthood and childhood BMI

Four separate GRSs were created using genome-wide significant SNPs from four European ancestry GWAS meta-analyses on reported age at menarche⁸ and age at voice breaking⁹, and measured BMI in adulthood (mostly middle-aged adults)²⁵ and

childhood (age range from 3 to 10 years)²⁶. Scores were calculated using 351 SNPs associated with age at menarche⁸, 73 SNPs associated with age at voice breaking⁹, 95 SNPs associated with adulthood BMI²⁵ (2/97 SNPs were not available in ALSPAC), and 15 SNPs associated with childhood BMI²⁶. The scores were constructed by multiplying the number of effect alleles (or probability of effect alleles if imputed) at each SNP (0, 1, or 2) by its weighting, summing them, and dividing by the total number of SNPs used, and reflect the average per-SNP effect on their respective trait (age at menarche, age at voice breaking, adulthood BMI, or childhood BMI). All scores were standardised (to mean=0 and SD=1) prior to analysis (**Supplemental Figure 2**).”

14. Lack of agreement between childhood BMI and Tanner: can it be due to the fact that the tanner scale is more challenging to implement on children with high BMI?

Response: This has been added as a possible explanation for this finding.

Page 20:

“In contrast to other pubertal indicators, higher childhood fat mass was associated with older (rather than younger) Tanner genitalia stage, and childhood BMI GRS was not associated with genitalia stage, which could both be due to Tanner staging being more challenging to implement in overweight or obese children²⁰, or because the more adipose children were more likely to exaggerate their development⁴⁸.”

Minor text edits

15. Most research relating to puberty timing has relied on reported age at menarche (a notable singular event, with no clear male equivalent) and therefore has been undertaken in females only. This seems to contradict the rest of the text and feels unspecific/wrong. Please consider rephrasing or removing this statement.

Response: We have removed this statement to better represent previous work, see our response to #1.

16. page 5 “age in age in Tanner pubic hair”. Please correct the repetition of “age in”

Response: Done.

17. this suit of nine measures. Suite?

Response: Thanks, this has been corrected (see Page 19).

18. In the discussion on the partial agreement between measures, I recommend also including <https://doi.org/10.1210/clinem/dgaa107>

Response: Thank you for the recommendation. We have included the results from this study in the Discussion.

Page 17-18:

“Our results agree with findings from the Edinburgh Longitudinal Growth Study (ELGS) where height, menarche, and clinical examinations of development stages

were taken every half-year until 20 years in 74 females and 103 males²⁹, and with a cross-sectional study of 703 Norwegian females aged 6-16 years that showed mean age in Tanner Stage 3 of breast and pubic hair development was younger than menarche³⁵.”

Reviewers' comments:

Reviewer #1 (Remarks to the Author):

I was Reviewer 1 for the previous submission and I appreciate the effort the authors have made in this revision. It has improved but I still find flaws that weaken the paper. I describe these below.

First, I will say that I rarely think it is appropriate for reviewers to make comments on the "paper that wasn't written." I will make one comment here. The data accessible to the authors is substantial. I find it frustrating that they minimize use of data by selecting single indicators. For instance, if they complete sequences of measures such as Tanner breast sequence, why choose only one point (stage 3). Variation in the initiation and progression of those markers would be more interesting. They present the overall data in the figures.

I previously mentioned that the term "Pubertal age" is confusing. In the revision the authors frequently use the term "Puberty age" which is even more confusing as "Puberty" is a noun and not a modifier. Also, my reason for saying either phrase is confusing is that puberty is not an event it is a period of change spanning several years. The current manuscript does not recognize this in the analysis and conclusions, and still discusses it as a milestone or singular point event.

The authors say they did not understand what I meant by "age-graded sample." The sample they use spans individuals at a range of ages. Indicators they use (e.g., Tanner staging) were created by evaluating similar samples and, where categorical measures are used, the resultant categories (i.e., stage 1, 2, 3, etc.) are age dependent. It is not surprising at all, then, that such indicators in a growing sample would be correlated, they are not independent of age. Similarly, as puberty is marked by dramatically increased values of a handful of hormones, it is not surprising that genetic correlations reflect this.

Conclusions include: "Findings from this prospective population-based cohort study of males and females supported all nine growth and development-based pubertal age measures as useful measures of age at puberty, by providing evidence that they are measuring the same biological process." This seems like circular logic to me. As puberty is defined by those measures (axillary hair, breast development, etc.) they are, of course, useful measures of age at puberty.

Also, in the last paragraph of the conclusions, they state: "Obtaining accurate measures requires repeat analyses whether using growth or developmental data, which can be challenging due to limited funding and research resources." I'm not really sure what this is referring to. They don't really discuss measurement accuracy (although it would be good if they did). Accuracy of measures of height, or weight, for instance, requires devices built for those measures (stadiometer, and scale) and does not require repeat analysis. The paragraph that follows is rambling and does not really address the conclusions of the paper.

The team of authors listed for this paper include notable names in growth analysis. It seems the first author is not adequately taking advantage of their input to the manuscript and that is unfortunate.

Reviewer #2 (Remarks to the Author):

The authors have answered all my comments.

I was Reviewer 1 for the previous submission and I appreciate the effort the authors have made in this revision. It has improved but I still find flaws that weaken the paper. I describe these below.

First, I will say that I rarely think it is appropriate for reviewers to make comments on the “paper that wasn’t written.” I will make one comment here. The data accessible to the authors is substantial. I find it frustrating that they minimize use of data by selecting single indicators. For instance, if they complete sequences of measures such as Tanner breast sequence, why choose only one point (stage 3). Variation in the initiation and progression of those markers would be more interesting. They present the overall data in the figures.

Response: We agree with the reviewer that we have access to a large amount of data that could be used for many additional analyses, including those suggested by the reviewer. The aim of this paper was to focus on pubertal timing. Pubertal timing estimates from growth models were based on the age at peak growth velocity (a measure commonly used to assess pubertal timing), which for the Tanner stages corresponds to stage 3. Variation in initiation and progression would be a nice follow-up paper but it’s beyond the remit of this paper. As the reviewer notes readers can see the data for all stages in the figure. No changes made.

I previously mentioned that the term “Pubertal age” is confusing. In the revision the authors frequently use the term “Puberty age” which is even more confusing as “Puberty” is a noun and not a modifier. Also, my reason for saying either phrase is confusing is that puberty is not an event it is a period of change spanning several years. The current manuscript does not recognize this in the analysis and conclusions, and still discusses it as a milestone or singular point event.

Response: We have used terminology that is most commonly used in research of pubertal timing. Whilst some may feel it is not grammatically correct it will be widely understood by a broad audience of readers from different backgrounds who are likely to be interested in the work. No changes made.

The authors say they did not understand what I meant by “age-graded sample.” The sample they use spans individuals at a range of ages. Indicators they use (e.g., Tanner staging) were created by evaluating similar samples and, where categorical measures are used, the resultant categories (i.e., stage 1, 2, 3, etc.) are age dependent. It is not surprising at all, then, that such indicators in a growing sample would be correlated, they are not independent of age. Similarly, as puberty is marked by dramatically increased values of a handful of hormones, it is not surprising that genetic correlations reflect this.

Response: The reviewer states that the correlations between stages are unsurprising – we agree. No changes made.

Conclusions include: “Findings from this prospective population-based cohort study of males and females supported all nine growth and development-based pubertal age measures as useful measures of age at puberty, by providing evidence that they are measuring the same biological process.” This seems like circular logic to me. As puberty is defined by those measures (axillary hair, breast development, etc.) they are, of course, useful measures of age at puberty.

Response: As the reviewer says, all the measures are useful in that they define a version of pubertal age. What the paper shows is that these different versions of pubertal age are consistent with each other, and hence support the idea that they are all measuring the same biological process. To make this clearer we have changed “useful” to “consistent”.

Also, in the last paragraph of the conclusions, they state: "Obtaining accurate measures requires repeat analyses whether using growth or developmental data, which can be challenging due to limited funding and research resources." I'm not really sure what this is referring to. They don't really discuss measurement accuracy (although it would be good if they did). Accuracy of measures of height, or weight, for instance, requires devices built for those measures (stadiometer, and scale) and does not require repeat analysis. The paragraph that follows is rambling and does not really address the conclusions of the paper.

Response: We have reworded the sentence: "Collecting longitudinal growth and development data can be challenging due to limited funding and research resources."

The team of authors listed for this paper include notable names in growth analysis. It seems the first author is not adequately taking advantage of their input to the manuscript and that is unfortunate.

Response: All authors have contributed to the paper in the ways described in the author contributions. Dr Elhakeem is an outstanding mid-career researcher, with evidence of expertise in growth modelling and pubertal timing demonstrated by invited lectures and PI grants in this area. He has led this project and worked appropriately with the co-authors. The more senior authors on this paper feel that reviewer comments about specific authors on a paper, which can only ever be assumptions, are inappropriate. In particular, comments like this could be damaging to early and mid-career researchers who are the future.